# RS-FairFRS: Communication Efficient Fair Federated Recommender System

## Abstract

Federated Recommender Systems (FRSs) aim to provide recommendations to clients in a distributed manner with privacy preservation. FRSs suffer from high communication costs due to the communication between the server and many clients. Some past literature on federated supervised learning shows that sampling clients randomly improve communication efficiency without jeopardizing accuracy. However, each user is considered a separate client in FRS and clients communicate only item gradients. Thus, incorporating random sampling and determining the number of clients to be sampled in each communication round to retain the model's accuracy in FRS becomes challenging. This paper provides sample complexity bounds on the number of clients that must be sampled in an FRS to preserve accuracy. Next, we consider the issue of demographic bias in FRS, quantified as the difference in the average error rates across different groups. Supervised learning algorithms mitigate the group bias by adding the fairness constraint in the training loss, which requires sharing protected attributes with the server. This is prohibited in a federated setting to ensure clients' privacy. We design RS-FairFRS, a Random Sampling based Fair Federated Recommender System, which trains to achieve a fair global model. In addition, it also trains local clients towards a fair global model to reduce demographic bias at the client level without the need to share their protected attributes. We empirically demonstrate across the two most popular real-world datasets (ML1M, ML100k) and different sensitive features (age and gender) that RS-FairFRS helps reduce communication cost and demographic bias with improved model accuracy.

## 1 Introduction

*Recommender systems* (RSs) have a wide variety of applications in online platforms like e-business, e-commerce, e-learning, e-tourism, music and movie recommendation engines (Lu et al. (2015)). Traditional RSs require gathering clients' private information at the central *server*, leading to serious privacy and security risks. ML models can train locally due to edge devices' increased storage and processing power. This has led to *Federated learning* (FL) (McMahan et al. (2017)), which allows clients to share their updates with the server without any data transfer. The server proposes a common model which is communicated with all clients. Using their data and the global model, clients train locally and communicate the updated model to the server. FL has found many applications in the past few years, e.g., Google keyboard query suggestion (Yang et al. (2018)), smartphone voice assistant, mobile edge computing, and visual object detection (Aledhari et al. (2020)). These applications face numerous challenges including communication efficiency (Smith et al. (2018)), statistical heterogeneity (Smith et al. (2017)), systems heterogeneity (Bonawitz et al. (2019)), privacy, personalization, fairness (Kairouz et al. (2021)), and many more. This paper focuses on two primary issues: communication efficiency and demographic bias in FRSs.

Unlike other applications of FL, where one client has data of many users, in FRS, each user acts as one client constituting a user's profile. FedRec (Lin et al. (2021)), an FRS, expects all the clients to train parallelly using *matrix factorization* (MF). In each communication round, the server aggregates the model updates from a huge number of local clients to obtain a global model, and this global model is then sent back to all the clients. This whole procedure increases the communication cost. We show that random sampling of clients in each communication round reduces the communication cost even when only item gradients of sampled users are communicated. Theoretically, we provide

bounds on an ideal fraction of clients to be sampled to maintain the model's accuracy. Proving sample complexity bounds is non-trivial as the clients may possess non-IID data. To circumvent this issue, we assume an underlying clustering structure on the clients such that clients within a cluster share similar item vectors. The main novelty lies in proving that the random sampling will fetch enough representation from each cluster and the predicted ratings obtained after sampling small number of clients will not be far (with high probability) from that of predicted ratings obtained after communicating with all clients in all the rounds.

Fairness in FRSs is a critical yet under-investigated area. Empirically, we prove that FedRec offers better recommendations to a particular group of clients. This unfair treatment can fortify the social stereotypes based on gender, race, or age, resulting in significant repercussions. So far, researchers have studied fairness in the domain of centralized RSs (Li et al. (2022)). Many past works (Islam et al. (2019); Yao & Huang (2017); Li et al. (2021); Yang et al. (2020a)) develop bias mitigation strategies in traditional RSs, which require sharing sensitive attributes with the server, causing privacy leakage in the federated setting. In FL framework, Yue et al. (2021); Kanaparthy et al. (2021); Du et al. (2020); Zhang et al. (2020) ameliorate bias in classification setting where each client possesses data of many users and thus can train for fairness in each communication round. As opposed to this, in FRS, each user acts as one client that sends its item gradients to the server after updating the user vectors and item gradients locally. This makes it extremely difficult to train locally towards fairness. We propose a dual-fair vector update technique with two phases. In Phase 1, the server aggregates the received item vectors and trains them towards fairness on a small fraction of data. Even if the global model is fair, local client updates may result in a heavily biased model. Thus in Phase 2, the clients minimize local error and learn item vectors closer to the globally fair vectors. In summary, our work aims at mitigating the issues of reducing the communication bottleneck and group bias in *Federated Recommendation system* (FedRec) (Yang et al. (2020b)) for the first time. We list down our main contributions below:

1. We provide sample complexity bounds on the fraction of clients required for maintaining accuracy within the desirable limit in Theorem4.1. Our experiments prove that sampling these many clients improve communication costs in FRS without affecting accuracy even when the clients do not disclose their user vectors and share only updated item gradients.

2. We show the existence of group bias in FRS quantified by evaluating discrepancies in the average group losses for each sensitive attribute. To mitigate this issue, we propose a novel *dual-fairness vector update* technique that tackles the issue of group fairness at local as well as global level.

3. Combining the ideas of random-sampling and dual-fairness vector update, we propose RS-FAIRFRS, a novel FRS model which provides communication efficiency and improved fairness as well as accuracy .

4. We show that RS-FAIRFRS mitigates demographic bias and improves accuracy via extensive experimentation on the two most popular datasets of ML1M and ML100K, with different demographics (age and gender).

## 2 RELATED WORK

We divide the literature review into four sections: (i) federated recommender systems (FRS), (ii) client sampling in federated learning, (iii) fairness in centralized RSs, and (iv) fair federated learning models. We emphasize that there does not exist any work which targets fairness in FRS.

**Federated Recommender Systems (FRS):** *Federated Collaborative Filtering* (FCF) (ud din et al. (2019)) is the first FRS to use implicit feedback for providing personalized recommendations. Lin et al. (2021) identifies the need for an FRS that uses explicit ratings and proposes FedRec that deploys two techniques- *Hybrid Filling* (HF) and *User Averaging* (UA) for privacy preservation. Researchers have been actively exploring many areas of research in FRS like denoising (Liang et al. (2021)), personalization (Jalalirad et al. (2019); Anelli et al. (2021)), privacy enhancement (Wang et al. (2021); Hu et al. (2021); Ali et al. (2021)), building robust FRS (Wu et al. (2022); Zhang et al. (2022); Rong et al. (2022)),mitigating cold-start issue (Wahab et al. (2022)), and improving accuracy of FRS (Perifanis & Efraimidis (2022)). FedFast (Muhammad et al. (2020)) speeds up training in FRS by using an active sampling technique based on the clustering of the user vectors.

However, Lin et al. (2021) argues that only item vectors of clients should be shared with the server to preserve clients' privacy. Further, Saito et al. (2019) explaints that usage of implicit feedback leads to the positive-unlabeled problem. Thus, we aim to analyze FedRec (Lin et al. (2021)) as our base model. It uses explicit feedback, unlike the other existing models like FCF (ud din et al. (2019)), NCF (Perifanis & Efraimidis (2022)), and Federank (Anelli et al. (2021)), which either user implicit feedback or convert explicit ratings to implicit feedback. Moreover, none of the aforementioned works investigates fairness in federated recommendations. We identify two major challenges of high communication costs and demographic bias, which remain uninvestigated. Since client sampling methods can significantly reduce communication costs, we now discuss some existing papers that use random sampling in FL.

**Client Sampling in Federated Learning:** Malinovsky et al. (2022) terms the selection of a batch of clients by the server for participation in the training process as *partial participation* and demonstrate that local steps can help to overcome the communication bottleneck. Charles et al. (2021) conducted extensive experiments to elicit significant challenges such as generalization issues, diminishing returns, training failures, and fairness concerns due to using large cohorts in FL models. Authors in (Balakrishnan et al. (2022)) adopt a greedy strategy to select clients to represent the overall population and provide convergence guarantees for the same. Fraboni et al. (2021a) used clustered sampling for better client representation and reduced variance of stochastic aggregation weight, and Chen et al. (2020) restricted the number of clients allowed to communicate their updates to the server. Finally, Fraboni et al. (2021b) studies the impact of client sampling on the convergence of FL models. While all the above methods assume unbiased client selection, another line of work by Cho et al. (2020) offers the first federated optimization convergence study for biased client selection techniques. Compared to classification settings, FRSs are different as out of the user, only item gradients are communicated to the server. None of the above papers provide bounds on the ideal number of samples required during communication even in federated learning setting. This paper, for the first time provides the sample complexity bounds with respect to FRSs.

**Group Fairness in RSs:** Xiao et al. (2017) study group fairness by maximizing the satisfaction of each group while minimizing the unfairness between them. Fu et al. (2020) propose a fairness constrained approach via heuristic re-ranking to mitigate group bias in explainable RSs. Li et al. (2021) categorizes clients into advantaged and disadvantaged groups according to their activity level and provides a re-ranking approach to debias the recommendations. Beutel et al. (2019) introduce novel metrics using pairwise comparisons to provide reasoning to bias and offer a regularizer to encourage improving the corresponding metric. Yao & Huang (2017) formalizes four novel metrics to quantify demographic bias and introduce a regularizer term in the objective function to mitigate demographic bias; this approach is somewhat similar to Padala & Gujar (2020). All these methods require the availability of sensitive features of clients to get fair recommendations, leading to privacy leakage in federated settings. A few fair approaches like (Edizel et al. (2020); Bobadilla et al. (2020)) do not require sensitive attributes for mitigating bias but learn disparity from data while training to get unbiased recommendations for the clients whose demographics are unknown. However, FedRec only permits sharing item vector updates with the server, thus making these techniques inapplicable to building fair FRS. Unlike all these approaches, our model RS-FAIRFRS uses an in-processing method that neither disturbs the original data nor requires information leakage to the server.

**Group Fairness in Federated Learning:** Papadaki et al. (2022) provides an optimization algorithm to improve group fairness with similar performance guarantees to centralized ML models. Kanaparthy et al. (2021) proposes four heuristics by considering balanced and heterogeneous data cases separately for fair federated classification models. Yue et al. (2021) propose GI-Fair to tackle group and individual fairness in FLSs by using a regularization term to penalize the spread in the aggregated loss. Recent work by Hu et al. (2022) uses the concept of bounded group loss to provide theoretical guarantees in group fairness. All these works were proposed to solve the issue of demographic fairness in a federated classification setting. They can not be applied to FRS as in FRS each user acts as one client, unlike classification, where one client can have data of many users. Unique from all other methods, RS-FAIRFRS is the first algorithm to provide dual-fairness vector updation in FRS by learning locally towards the global fair model for local fairness and achieving global fairness by training aggregated vectors towards fairness.

## 3 PRELIMINARIES

**FedRec:** In a typical FRS with explicit feedback, we have $n$ users (or clients), $u \in \{1, 2, 3, ......., n\}$ and $m$ items, $i \in \{1, 2, 3, ........, m\}$. Each client $u$ has it's rating vector $[r_{ui}]_{i=1}^m$ that depicts the rating given by a client $u$ to an item $i$. The true ratings given by the user and predicted ratings are denoted using $r_{ui}$ and $\hat{r}_{ui}$, respectively. We assume that $r_{ui} = 0$ if the client has not rated an item. $p_{ui} \in \{0, 1\}$ acts as an indicator variable for rated and unrated items. FedRec uses matrix factorization that identifies the latent structure behind the data to generate two matrices $U \in \mathbb{R}^{n \times k}$ and $V \in \mathbb{R}^{m \times k}$ in a way that each client $u$ is associated with a vector $U_u \in \mathbb{R}^{1 \times k}$, called as *user vector* and each item $i$ is associated with a vector $V_i \in \mathbb{R}^{1 \times k}$, termed as *item vector*. The predicted rating of $i^{th}$ item by $u^{th}$ user can be computed as $\hat{r}_{ui} = U_u.V_i^T$. The goal of FedRec is then to learn the user vectors (locally) and item vectors globally to minimize the loss function:

$$\mathcal{L}^{\mathcal{MF}} = \sum_{u \in [n]} \sum_{i \in [m]} p_{ui}(r_{ui} - U_u.V_i^T)^2 + \lambda^r(\| V_i \|^2 + \| U_u \|^2) \tag{1}$$

FedRec aims at predicting the rating of a client $u$ for each item $i$ without sharing their rating behaviors or records. For this, some unrated items are randomly sampled using sampling parameter $\rho$ and assigned virtual rating. Then, item gradients $\nabla V(u, i)$ for both truly as well as virtually rated items are shared with the server. The server than aggregates these gradients and sends back the aggregated item vectors to all the clients. Additionally, each client also computes the user gradient $\nabla U_u$ locally and is not shared with the server. Next section proposes RS-FAIRFRS that solves the issue of communication inefficiency in FRS and mitigates demographic bias.

## 4 PROPOSED METHODOLOGY

To reduce the communication cost, we randomly sample clients in each communication round who communicate $\nabla V(u, i)$ with the server. Randomly sampling the clients has been proposed in literature under supervised federated learning (Charles et al. (2021); Fraboni et al. (2021b); Cho et al. (2020)). However, since each client is a separate user in FRS and user only shares the item gradients with the server, it is not clear if random sampling will aid the recommender system to reduce communication cost without affecting its accuracy in a federated setting. More importantly, the main question is that how many clients should we sample in each communication round. To answer this question, next section provides sample complexity bounds on the number of clients required to be sampled to obtained the desired accuracy.

### 4.1 RANDOM SAMPLING WITHOUT REPLACEMENT

Unlike FedRec (Lin et al. (2021)), where the server aggregates item vectors after all the clients have sent their updates, in RS-FAIRFRS, the server uniformly samples a $\tau$ fraction of $n$ clients. It is well known that the users in FL are non-IID. However, users who provide ratings to items tend to possess an underlying clustered structure. Various algorithms (Koren et al. (2009); Gupta et al. (2020)) work by identifying the latent patterns of users to provide recommendations and within the cluster, users are IID. We aim to utilize this homogeneity within the same clusters without the knowledge of the $K$ clusters and the users belonging to each cluster. For random sampling to work, it is important that the sampled set of clients $C^\tau$ must represent the entire population. In the first result, we show that when certain number of clients are sampled randomly at uniform, they represent each cluster equally to ensure that this sampled set is enough to represent the entire population.

**Lemma 1** *Suppose $n$ clients are uniformly distributed amongst $K$ clusters. Then, a subset $S \subseteq [n]$ sampled uniformly at random (without replacement) will contain an approximately equal number of clients from each cluster.*

We use Hoeffding's bound (Serfling (1974)), which holds for sampling without replacement but provides a very loose bound. Let $X_i^j \in \{0, 1\}$ denote the random variable taking the value 1 when $i^{th}$ sample belong to cluster $j$ and 0 othersise. Then using Hoeffding's bound, we get $\mathbb{P}\left(| \sum_i X_i^j - \frac{|S|}{K}| \geq \epsilon\right) \leq 2 \exp\left\{\frac{-2\epsilon^2}{|S|}\right\}$. If we take $|C^\tau| = 2100$, i.e. 35% of the total number of clients (6000), then we get this probability to be roughly around 0.62, which is obtained at $K = 10$.

Hoeffding's inequality provides a very loose bound but actually this probability is very high which is evident from some basic experiments provided in the Appendix.

**Lemma 2** *Given $n$ clients during the training, $\tau$ represents the fraction of clients sampled for each communication round. If $\bar{V}_i^{\tau} = \frac{1}{n\tau} \sum_{i \in C^{\tau}} V_i$ denote the average of item vectors over some $C^{\tau}$ clients and $\bar{V}_i^n = \frac{1}{n} \sum_{i=1}^{n} V_i$ represent the average of item vectors over total $n$ clients. Then, $\mathbb{E}[U_u^T \bar{V}_i^{\tau}] = \mathbb{E}[U_u^T \bar{V}_i^n]$*

Here, $U_u^T \bar{V}_i^{\tau}$ and $U_u^T \bar{V}_i^n$ denotes the predicted rating of any user $u$ when aggregated item vectors are obtained only via the sampled users and all the clients in the training set respectively. This lemma holds inherently true if the underlying clients are homogeneous which is not true in FL. Thus we use the latent clustering assumption and Lemma 1 to prove that the expected values of predicted ratings of sampled as well as the entire population are equal. The detailed proof is provided in Appendix.

Now, we state our main theorem below:

**Theorem 4.1 (Random Sampling of Clients)** *Given a rating matrix $R$, let $< \{U_u\}_{u=1}^{n}, \{V_i\}_{i=1}^{m} >$ denote a Federated Recommendation Model with predicted ratings lying within a range of $[a, b]$. If $\bar{V}_i^{\tau} = \frac{1}{n\tau} \sum_{i \in C^{\tau}} V_i$ and $\bar{V}_i^n = \frac{1}{n} \sum_{i=1}^{n} V_i$ represent the average of item vectors over some $\tau$ fraction of clients and total $n$ clients, respectively, then*

$$\mathbb{P}(|U_u^T \bar{V}_i^{\tau} - U_u^T \bar{V}_i^n| \geq \epsilon) \leq 2 \exp \left\{ \frac{-n\tau\epsilon^2}{2(b-a)^2} \right\}$$

The above theorem can be proved using Hoeffding's bound and Lemma 2 According to the above theorem, if the ratings lie between $[1, 5]$, then the probability that the error in predicted rating is more than $10\%$ is less than $1\%$ with just $35\%$ clients from the pool of 6000 clients. Therefore, from our theorem if a dataset has around 6000 clients, we chose $\tau = 0.35$ in our experiments.

It is important to note that our main contribution lies in showing that while bounding the sample complexity, in general, is a hard problem, clustering assumption on the underlying data makes it possible to provide the non-trivial bounds. Without this assumption, straightforward use of Hoeffding's inequality will give trivial bounds of $100\%$ on the sample complexity, whereas we need only $30\%$. It is important to note that we assume the existence of clusters of item vectors with almost equal number of vectors in each cluster for theoretical analysis. We prove the same experimentally in Appendix. Furthermore, the clustering only aids in acquiring a bound on ideal fraction of clients to be sampled. The fairness of RS-FAIRFRSis independent of any clustering with or without groups (age/gender). Since private clustering is an open challenge in FL and forming clusters require sharing of sensitive attributes, RS-FAIRFRSensures privacy to most of its users by hiding their demographics from the server.

## 4.2 DUAL-FAIR UPDATE

This section firstly discusses the inability of existing fairness notions in RSs to be able to measure group bias in a federated setting. We then propose fairness metric which when added as a constraint in optimization function at server helps achieve a fair global model. Furthermomre, we discuss a two phase mechanism which helps in achieving global as well as local fairness.

Existing fairness notions in RS, namely value unfairness, absolute unfairness, and non-parity fairness (Yao & Huang (2017)) consider the difference in average loss on a specific item $i$ concerning users belonging to advantaged and disadvantaged groups. It is not feasible in FL as each client is distinct and prohibited from sharing its ratings with the server and other clients. Moreover, Mansoury et al. (2020) mentions profile size as an important factor in group bias as more active clients receive better recommendations. We define *user activeness* as the number of items rated by a user, i.e. a user is more active if he or she has rated more items $I_u$ out of all the items in the item set $I$. Thus, with an assumption of a binary attribute, our metric *accuracy parity* ($\mathcal{L}^{ap}$) normalizes the sum of squared loss over all the items rated by a client and is given as:

$$\mathcal{L}^{ap} = \frac{1}{|g|} \sum_{u \in g} \frac{1}{|I_u|} \sum_{I \in I_u} (\hat{r}_{ui} - r_{ui})^2 - \frac{1}{|\neg g|} \sum_{u \in \neg g} \frac{1}{|I_u|} \sum_{I \in I_u} (\hat{r}_{ui} - r_{ui})^2 \tag{2}$$

Here, $g$ and $\neg g$ denote the disadvantaged and advantaged groups, respectively. $I_u$ is the set of items rated by $u$ and $\hat{r}_{ui}, r_{ui}$ depict the predicted and true ratings, respectively. For models $\theta$ and $\bar{\theta}$, we

say that $\theta$ is a demographically fairer model if $\mathcal{L}^{ap}(\theta) < \mathcal{L}^{ap}(\bar{\theta})$. Designing a fair FRS involves tackling three significant challenges which must be tackled. (i)Designing debiasing techniques for local clients becomes challenging if they are reluctant to share their sensitive attributes (Ezzeldin et al. (2021)).(ii) Considering the settings where each user is not one client, local fairness does not ensure global fairness in FL. The non-IID data in FL makes it impossible for the entire distribution to be represented by one standard distribution.(iii) To achieve fairness, simple federated models usually do weighted aggregation; however, it becomes infeasible to assign weights to the updates sent to the server without knowing the group to which they belong. Dual-Fair Updation involves training towards fairness in two phases described below.

**FairMF:** We assume the availability of data of very few (20%) clients $D_{server}$ at the server for evaluating the fairness loss.This assumption was also considered by Kanaparthy et al. (2021) for building a fair federated model for face classification. Previous works considered 5% of the overall data on the server. Choosing 5% of the overall data in RSs may lead to the privacy leakage of more users. Hence, instead of selecting 5% of the entire population, we select data of only 20% of the users. This assumption helps achieve fairness without expecting clients to reveal their private sensitive attributes to the server even during training. Therefore, unlike a simple FRS, the server in RS-FAIRFRS not only aggregates but also helps achieve fairness. We denote each client at the server as $s \in \{1, ...., S\}$ such that $S << n$, i.e. the number of clients at the server will always be much less than the number of local clients. FairMF trains the data at server $D_{server}$ for obtaining global fairness objective. The goal of FairMF is to optimize the loss function defined as the combination of regularized MF (equation 1) and fairness penalty (equation 2). The final loss function is

$$\min_{U,V} \mathcal{L}^{MF} + \lambda^f \mathcal{L}^{ap} \tag{3}$$

The hyperparameter $\lambda^f$ acts as a fairness penalizer. The update equations for client vector $U_u$ and item vector $V_i$ are obtained by taking derivatives of the fairness loss function with respect to client and item vectors respectively. Server runs FairMF for some iterations $t_s$ and obtains final $U_{fair}$ and $V_{fair}$. Finally, $V_{fair}$ and $[V_i]_{i=1}^m$ (aggregated item vectors) are communicated to all the clients. We provide the exact procedure for FairMF in Algorithm 1.

**FO-ClientBatch:** *Fairness Oriented Client Batch* ensures local fairness by learning locally towards fair global model. Each client downloads both $V_{fair}$ and $V_i$ from the server. While $V_{fair}$ contributes towards fairness, communication of aggregated $V_i$ provide each user with the benefit of other users' participation. Since in FL models global fairness does not ensure local fairness, it is important that each local client also trains towards the fair model. Then, each user vector is updated locally, followed by an updation in the $V_i$ towards $V_{fair}$. Thus the local objective function changes to

$$\min_{U,V} \mathcal{L}^{MF} + \eta(||V_{fair} - V_i||^2) \tag{4}$$

Clients keep $\nabla U_u$ with themselves and communicate only $\nabla V$ to the server. The section below describes the detailed algorithm and communication details.

---

**Algorithm 1** FairMF $(D_{server}, U_{fair}, V)$

1: **for** $T_s = 1, 2, ...., t_s$ **do**
2:     **for** each $(s, i)$ in $D_{server}$ **do**
3:         if $r_{ui} \neq 0$, update $U_s$ and $V_i$ by calculating gradients by differentiating equation equation 3
4:     **end for**
5: **end for**
6: $U_{fair} \leftarrow U_s, V_{fair} \leftarrow V_i$
7: **return** $U_{fair}, V_{fair}$

---

**Algorithm 2** ClientFilling( $V_i, i = 1, 2, ...., m; U_u; u; t$)

1: **if** strategy == HF **then**
2:     **for** $t_{local} = 1, 2, ....., T_{local}$ **do**
3:         Calculate the gradient $\nabla U_u$
4:         $U_u \leftarrow U_u - \gamma \nabla U_u$
5:     **end for**
6:     Assign $r'_{ui}$ to each item $i \in I'_u$ via HF
7: **else**
8:     Assign $r'_{ui}$ to each item $i \in I'_u$ via UA
9: **end if**

---

### 4.3 RS-FAIRFRS

Algorithm3 describes RS-FAIRFRS in whole. With the assumption of the availability of a small dataset ($D_{server}$) corresponding to $s$ users at the server, the procedure begins by initializing $V = [V_i]_{i=1}^m$ and $U = [U_u]_{u=1}^s$ which represent the item and user vectors respectively. Then, some $\tau$

---

**Algorithm 3** RS-FairFedRec Communication Efficient and Fairness Aware Federated Recommender System

---

**Input:** $D_{server}, D_{train}, \tau, \gamma, \lambda^r, \lambda^f, \alpha, \rho$
**Output:** $U_u, V_i$
1: Initialize $V = [V_i]_{i=1}^m$ and $U = [U_u]_{u=1}^s$
2: **for** $t = 1, 2, ......, T$ **do**
3:     $C^\tau \leftarrow \{$Randomly sub-sample $\tau$ fraction of clients from all the users in $D_{train}\}$
4:     $U_{fair}, V_{fair} \leftarrow$ FairMF$(D_{server}, V, U_{fair})$
5:     **for** each client $u \in U$ in parallel **do**
6:         Sample $I'_u$ from $I \setminus I_u$ with $|I'_u| = \rho|I_u|$
7:         ClientFilling$(V_i, i = 1, 2, ...., m; U_u; u, t)$
8:         Calculate the gradients $\nabla U_u$ and $\nabla V(u, i)$ by differentiating equation 3
9:         Update $U_u$ via $U_u \leftarrow U_u - \gamma \nabla U_u$
10:    **end for**
11:    **for** i = 1,.....,m **do**
12:        Calculate the aggregated gradient $\nabla V_i$ for clients in $C^\tau$
13:        Update $V_i = V_i - \gamma \nabla V_i$
14:    **end for**
15:    Decrease the learning rate $\gamma \leftarrow 0.9\gamma$
16: **end for**

---

fraction of total clients present in the *training dataset* ($D_{train}$) are randomly sampled by the server denoted by $C^\tau$. The assumption of $D_{server}$ helps in obtaining fair item vectors $V_{fair}$ which are communicated to all the clients for local fairness. Alongwith these, $V$ is also sent to each client to retain the federated properties of FedRec and allow benefits of the participation of other clients. For initial round, the initialized item vectors are communicated, however as the training preoceeds, $V$ gets updated by the aggregated item vectors. Further, the server aggregates item vectors sent by $C^\tau$ clients only. The fair item vectors $V_{fair}$ and aggregated item vectors communicated by the server are received by each client $u$ and local training happen at all the clients. Similar to FedRec, using UA, every client will receive some ($\rho$) virtual ratings. In UA, some unrated items are sampled and then a virtual rating $r'_{ui}$ is assigned to these items. In HF, after a certain amount of time $T_{predict}$ the virtual rating is replaced by the predicted rating. To acquire predicted rating each client evaluates its user gradients and then updates its user vector. This procedure is called Client Filling (Lin et al. (2021)) explained in Algorithm 2. The user and item gradients are evaluated simply by differentiating the equation 4. The proposed local objective minimizes the squared loss between the global item vectors and local ones. The updated item gradients are uploaded to the server to train FairMF on the $D_{server}$ and obtains $U_{fair}$ and $V_{fair}$. This procedure is repeated till convergence.

## 5 EXPERIMENTS

We empirically show that (i) random sub-sampling of clients reduces communication costs without any slump in accuracy of an FRS , (ii) FedRec suffers heavily from demographic bias, and (iii) RS-FAIRFRS significantly reduces bias without any leakage of client's privacy and improves accuracy.

### 5.1 EXPERIMENTAL SET-UP

We use two benchmark datasets *ML1M* [1] and *ML100K* [2] with explicit ratings ranging from 1 to 5. *ML1M* dataset consists of $1,000,209$ ratings of $3,706$ movies by $6,040$ users and *ML100K* consists of $943$ users with $100,000$ ratings for $1,682$ items. ML1M has $4,331$ males and $1,709$ females; also, there are $5,818$ people with age $> 18$ and $212$ users with age $< 18$. Similarly, in ML100k, users with age above 18 ($889$) are much more than ones with under 18 ($54$). Furthermore, ML100k has only 273 females as compared to 670 males. All our experiments use RMSE (Root Mean Square Error) as used by Lin et al. (2021) to provide a fair comparison of the accuracy of our model with

---

[1]https://grouplens.org/datasets/movielens/1m/
[2]https://grouplens.org/datasets/movielens/100k/

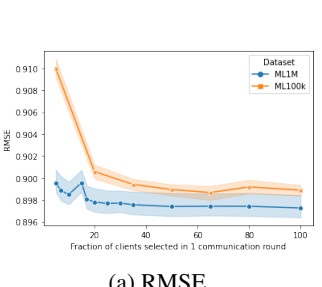 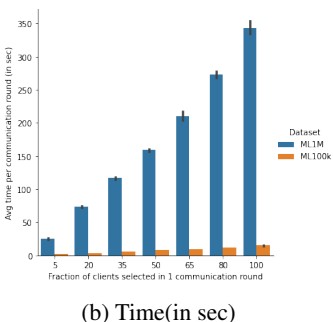

(a) RMSE        (b) Time(in sec)

Figure 1: Comparison plots for accuracy and average time per communication round on different values of $\tau$ in RS-FedRec on two datasets of ML1M and ML100k

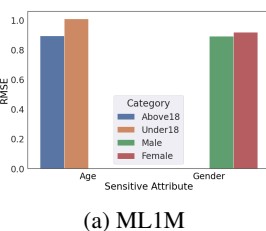 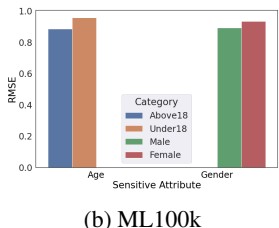

(a) ML1M        (b) ML100k

Figure 2: Difference in RMSE scores for different sensitive attributes by FedRec over two datasets.

the FedRec. We also provide an analysis of Group Losses, calculated using the average sum of the RMSE score of all clients belonging to a certain group. Finally, we use $\mathcal{L}^{ap}$ [equation 2] to evaluate demographic bias in all the experiments. We split each dataset randomly by keeping $20\%$ of the data for the test set and the rest for training. Similar to Lin et al. (2021), we use $k = 20$ latent features and set the value of sampling parameter to $\rho = 2$. We run all the models till convergence ($T = 20$) and the FairMF procedure is executed to $t_s = 15$ runs. We report all our results over an average of 10 runs. The final tuned hyperparameter values and other details are provided in the Appendix. While it can be apparent that more data on the server will yield better performance, we run all our experiments on RS-FAIRFRS with $100\%$ and RS-FAIRFRS with $20\%$ data on the server to show that even with all the data on the server, RS-FAIRFRS with only a $20\%$ data can provide much better accuracy as well as fairness. For exact numbers of the training and testing results, refer Appendix. Now we provide interesting results via various graphical representations.

## 5.2 EMPIRICAL EVALUATION

**Random Sampling of Clients without Replacement:** We randomly sample $\tau$ fraction of all the available clients. The model's accuracy is analyzed for a wide range of $\tau$ values in Figure 1a. Our model has a very high error for small $\tau$ values. This indicates the failure to converge and with around $35\%$ of clients, the model performs nearly the same as with $100\%$ clients. From Theorem4.1, it can be shown that the predicted rating of each client with sampled item vectors will be within $5\%$ error with a probability of at least 0.96. In line with the findings of Charles et al. (2021), increasing $\tau$ increases the training loss. Figure 1b presents the average time taken (in sec) for one communication round by FedRec with different values of $\tau$. From these inferences, for all our experiments we select $\tau = 35$ as an ideal value which provides reasonable accuracy in very less time.

**Disparate treatment of FedRec for certain sensitive attributes:** To demonstrate the biased treatment of FedRec towards certain sensitive groups, we compute average losses on each sensitive attribute. Figure 2 shows that the groups with more number of clients *(advantaged groups)* enjoy more accurate recommendations where as ones with lesser number of clients *(disadvantaged groups)* receive more erroneous recommendations. In both the datasets, clients with age $> 18$ enjoy recommendations with lesser RMSE as compared to the ones with age $< 18$. Similarly, females of both the datasets obtain recommendations with much higher RMSE as compared to the males. This shows

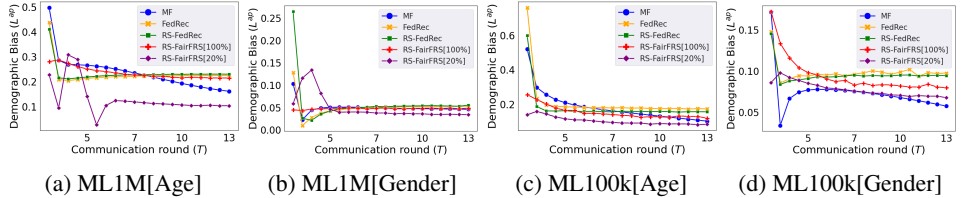

Figure 3: $\mathcal{L}^{ap}$ on two datasets and two different demographics for different algorithms.

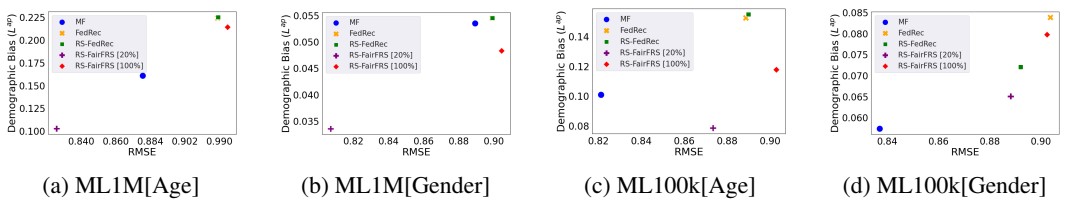

Figure 4: Fairness vs Accuracy plots

that the existing class imbalance affects the recommendations provided by FedRec which motivates us to develop a fair FedRec that trains towards mitigating this disparate treatment of FedRec.

**Fairness of RS-FAIRFRS:** To analyze fairness of RS-FAIRFRS, we compare our results with FedRec, RS-FedRec (FedRec with sampling) and RS-FAIRFRS with (100%) data at the server. We also provide bias results on MF to show that FedRec amplifies the bias due to aggregation. In Figure 3, we show that only (20%) of client's data at the server helps in achieving better fairness than 100% because our model uses random sampling of clients in each round. The sampled clients make use of the server dataset to obtain fair results. 100% data on the server includes many outliers which worsens the fairness of sampled clients thus generating poor fairness as well as accuracy. Our results in Appendix show that even with only 25% and 50% of the items at the server, our model is able to reduce bias. Further, it is important to note that our model uses randomly initialized user and item vectors. Slowly as the training proceeds towards minima, the random vectors get trained and updated to obtain closest rating predictions. Thus, over a period of time, the curve smoothens but the initial readings can be very fluctuating. It is evident from all the four graphs in Figure 3 that RS-FAIRFRS with 20% data on the server manages to acquire least bias on both the datasets for all the sensitive attributes. Thus, RS-FAIRFRS(20%) manages to significantly reduce the bias in FedRec without leaking any sensitive information of most of the users during training.

**Fairness vs Accuracy in RS-FAIRFRS:** RS-FAIRFRS improves group bias and generates more accurate recommendations. We compute RMSE as well as $\mathcal{L}^{ap}$ to depict the excellence of our model in both the aspects. Figure 4 shows that RS-FAIRFRS(20%) provides more fair and accurate recommendations to the clients in both datasets for both the genders as well as users with age above and under 18. It can be easily inferred that a lower value in both Demographic bias and RMSE accounts for better model. Users belonging to both the age groups in ML100k dataset enjoy fair recommendations with RS-FAIRFRS(20%). Though, MF performs slightly better in terms of accuracy as well as fairness for users in ML100k for gender attribute due to the efficiency of centralized settings for smaller datasets, overall, RS-FAIRFRS(20%) outperforms all the existing federated algorithms.

## 6 CONCLUSION AND FUTURE WORK

We propose RS-FAIRFRS which incorporates two key ideas (i) random sampling of clients to reduce the communication cost, and (ii) dual-fair updation to mitigate group bias. We are first to theoretically bound the sample complexity on the fraction of clients to be sampled without affecting model's accuracy. We empirically demonstrate our theoretical results and provide extensive experiments to show that we can achieve much-reduced bias and improved model accuracy. Exploring various other types of biases in FRSs can be an interesting research direction. We would also like to extend our analysis to provide theoretical fairness guarantees of RS-FAIRFRS and explore the usage of dual-fair updation-like techniques in many other FL domains in future.

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

# A    APPENDIX

## A.1    NOTATIONS

We list down all the important and frequently used notations in Table 1.

| Symbol | Meaning |
|--------|---------|
| $n$ | Number of clients in training set |
| $m$ | Number of items in training set |
| $K$ | Number of Latent factors |
| $R = \{1, ......5\}$ | Rating range |
| $r_{ui}$ | True rating |
| $\hat{r}_{ui}$ | Predicted rating |
| $r'_{ui}$ | Virtual rating |
| $p_{ui}$ | Indiactor variable |
| $u$ | client |
| $i$ | item |
| $I$ | Set of all items |
| $U_i$ | Set of clients who rated item $i$ |
| $U'_i$ | Set of all clients who rated an item $i$ virtually |
| $I_u$ | Set of items rated by $u$ |
| $I'_u$ | Set of items rated virtually by $u$ |
| $U$ | client vectors matrix |
| $V$ | Item vectors matrix |
| $U_u$ | client vector for client $u$ |
| $V_i$ | Item vector for an item $i$ |
| $\nabla V(u, i)$ | Item gradient for $i^{th}$ item which was rated by a user $u$ |
| $R$ | Rating Dataset |
| $D_{train}$ | Training set |
| $D_{server}$ | Dataset at server |
| $g$ | Disadvantaged Group |
| $\neg g$ | Advantaged Group |
| $\rho$ | Sampling parameter |
| $\lambda^r$ | Regularizer to prevent overfitting in FairMF |
| $\lambda^f$ | Fairness regularizer in FairMF |
| $\alpha$ | Learning rate in FairMF |
| $\tau$ | Fraction of clients sampled for one communication round |
| $\gamma$ | Learning Rate for clients |
| $\eta$ | Fairness Penalizer for clients |
| $T$ | Number of communication rounds |
| $T_s$ | Number of epochs in FairMF |
| $T_{local}$ | Number of iterations required to obtain prediction in Client Filling |

Table 1: Notations and their meaning

## A.2    EXPERIMENTAL PROOF FOR LEMMA 1

This section provides the experimental proof for Lemma 1. FedRec generates item vectors at each communication round. We cluster these item vectors generated by FedRec by using K-means clus-

tering (Bock (2007)). Figure 5 shows the elbow curve obtained after plotting clustering loss *Inertia* (Hedman et al. (2007)) corresponding to the number of clusters. Inertia is one of the most populalr metrics to evaluated clustering results. It is calculated as the sum of distances to each cluster center. A high value of total distance implies that the points are far from each other which is a clear indication that they are less similar to each other. It can be seen that for both the datasets we get $K = 20$ as the ideal number of clusters.

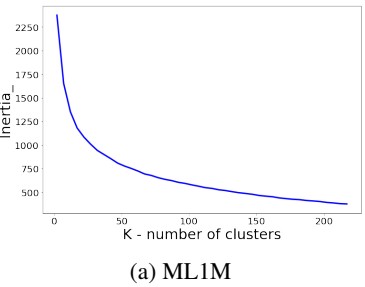 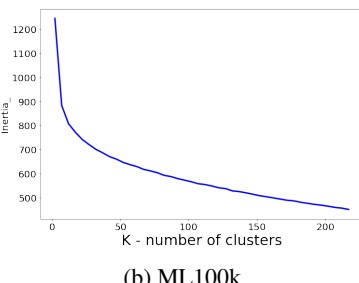

(a) ML1M  (b) ML100k

Figure 5: Ideal number of clusters for item vectors in both datasets

Considering $K = 20$ and asssuming that we have total number of clients $n = 6000$, we randomly assign each client to one out of 20 clusters. Then, we randomly sample some of the clients and observe some interesting observations depicted in Figure 6. Figure 6a shows that after sampling some clients randomly at uniform, the average number of clients from each cluster are almost same. Further, Figure 6b clearly shows that the probability of getting $> 15\%$ error in above experiment is as small as $7.5\%$ when averaged over 500 runs. Finally, Figure 6c demonstrates that minimum number of samples in each cluster is also same. Thus all the three experiments support our Lemma 1.

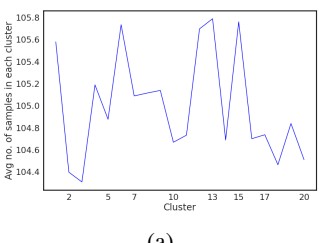 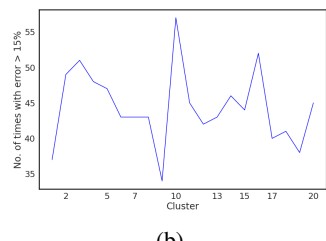 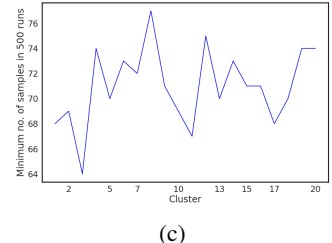

(a)  (b)  (c)

Figure 6: Experimental Analysis of Random Sampling of Clients and Clustering

### A.3 THEORETICAL ANALYSIS OF CLIENT SAMPLING

We first recall Lemma 2 and then provides its proof below:

**Lemma 2** *Given $n$ clients during the training, $\tau$ represents the fraction of clients sampled for each communication round. If $\bar{V}_i^\tau = \frac{1}{n\tau}\sum_{i \in C^\tau} V_i$ denote the average of item vectors over some $C^\tau$ clients and $\bar{V}_i^n = \frac{1}{n}\sum_{i=1}^n V_i$ represent the average of item vectors over total $n$ clients. Then, $\mathbb{E}[U_u^T \bar{V}_i^\tau] = \mathbb{E}[U_u^T \bar{V}_i^n]$*

**Proof:** Let $S_j$ denote the set of clients in cluster $j$, and $C_j$ denote the set of clients sampled from cluster $j$. Since in each cluster, the item vectors are coming from identical distribution, we have $\mathbb{E}[U_u^T \bar{V}_i^{S_i}] = \mathbb{E}[U_u^T \bar{V}_i^{C_i}]$. Here, $\bar{V}_i^{S_i}, \bar{V}_i^{C_i}$ represent the average of item vectors $V_i$'s from sets $S_i$ and $C_i$ respectively. From Lemma 1, if $\tau$ fraction of clients are selected from $n$ clients uniformly at random without repetition, then we have $C_j \approx \tau S_j$ with high probability. Then,

$$\mathbb{E}[U_u^T \bar{V}_i^\tau] = \frac{1}{\sum_{j=1}^l |C_i|}\sum_{j=1}^l |C_i|\mathbb{E}[U_u^T \bar{V}_i^{C_i}]$$

$$= \frac{1}{\sum_{j=1}^{l} \tau |S_i|} \sum_{j=1}^{l} \tau |S_i| \mathbb{E}[U_u^T \bar{V}_i^{S_i}]$$

$$= \mathbb{E}[U_u^T \bar{V}_i^n]$$

We now recall our main Theorem 4.1 and provide its detailed proof using above Lemma and Hoeffding's bound.

**Theorem 4.1 (Random Sampling of Clients)** *Given a rating matrix R, let $< \{U_u\}_{u=1}^{n}, \{V_i\}_{i=1}^{m} >$ denote a Federated Recommendation Model with predicted ratings lying within a range of $[a, b]$. If $\bar{V}_i^\tau = \frac{1}{n\tau} \sum_{i \in C^\tau} V_i$ and $\bar{V}_i^n = \frac{1}{n} \sum_{i=1}^{n} V_i$ represent the average of item vectors over some $\tau$ fraction of clients and total $n$ clients, respectively, then*

$$\mathbb{P}(|U_u^T \bar{V}_i^\tau - U_u^T \bar{V}_i^n| \geq \epsilon) \leq 2 \exp\left\{ \frac{-n\tau\epsilon^2}{2(b-a)^2} \right\}$$

**Proof of Theorem 4.1:** From Hoeffding's inequality, we have,
$\mathbb{P}(|U_u^T \bar{V}_i^\tau - \mathbb{E}[U_u^T \bar{V}_i^\tau]| \geq \epsilon) \leq 2 \exp\{\frac{-2n\tau\epsilon^2}{(b-a)^2}\}$, and
$\mathbb{P}(|U_u^T \bar{V}_i^n - \mathbb{E}[U_u^T \bar{V}_i^n]| \geq \epsilon) \leq 2 \exp\{\frac{-2n\epsilon^2}{(b-a)^2}\}$

Thus with probability atleast $1 - 2 \exp\{\frac{-2n\tau\epsilon^2}{(b-a)^2}\}$, we have,

$$\mathbb{E}[U_u^T \bar{V}_i^\tau] - \epsilon \leq U_u^T \bar{V}_i^\tau \leq \mathbb{E}[U_u^T \bar{V}_i^\tau] + \epsilon$$

$$\implies \mathbb{E}[U_u^T \bar{V}_i^n] - \epsilon \leq U_u^T \bar{V}_i^\tau \leq \mathbb{E}[U_u^T \bar{V}_i^n] + \epsilon$$

(From Lemma 2)

$$\implies U_u^T \bar{V}_i^n - 2\epsilon \leq U_u^T \bar{V}_i^\tau \leq U_u^T \bar{V}_i^n + 2\epsilon$$

Thus, we get $\mathbb{P}(|U_u^T \bar{V}_i^\tau - U_u^T \bar{V}_i^n| \geq \epsilon) \leq 2 \exp\{\frac{-n\tau\epsilon^2}{2(b-a)^2}\}$.

## A.4 HYPERPARAMETER TUNING

We carefully tune all the hyper-parameters and list down the final tuned values in Table 2. It is important to note that server in FedRec and RS-FedRec acts as an aggregator. Thus, the server does not require any tuning and the final tuned values for both the algorithms account for client side parameters. Opposite to this, server in RS-FAIRFRS[100%] as well as RS-FAIRFRS[20%] require fairness oriented training. Therefore, we present separate hyper-parameter values for client side and server side. Since, regularizers at server as well as client side prevents overfitting, we use $\lambda^r$ to denote them, but fairness regularizer at server $\lambda^f$ trains aggregated item vectors towards fairness using $D_s erver$ and fairness regularizer $\eta$ at client side trains local models towards globally fair item vectors. Experimentally, we observe that all the models converge well at $T = 20$, thus we use $T = 20$ in all our experiments. However, while executing FairMF at the server, we train FairMF for $T_s = 15$ after each communication round. We obtain better fairness when we train FairMF for some epochs instead of letting it converge fully over each communication round which can be time consuming and it increases the load at server too. Thus FairMF executes for only 15 epochs and provides the updated item vectors to be communicated to all the local clients.

## A.5 TRAINING AND TESTING RESULTS

We report the train as well as test results of all the four federated algorithms on two real-world datasets of ML1M and ML100k with two different sensitive attributes in Tables 3 and 4. It is noted that similar to training results, RS-FAIRFRS[20%] performs the best in terms of accuracy as well as fairness. From the tables, it is evident that FedRec and RS-FedRec performs nearly the same which shows that random sampling retains the model accuracy. Then, our results (rows for RS-FAIRFRS[20%] and RS-FAIRFRS[100%]) provide an empirical proof to the claim we made in the main paper which states that we do not need huge amount of data at the server. With a minimal

| Algorithm | | Hyper-parameter | Tuned Values | |
|---|---|---|---|---|
| | | | ML1M | ML100k |
| FedRec | | Learning Rate ($\gamma$) | 0.1 | 0.08 |
| | | Regularizer ($\lambda^r$) | 0.02 | 0.01 |
| RS-FedRec | | Learning Rate ($\gamma$) | 0.1 | 0.08 |
| | | Regularizer ($\lambda^r$) | 0.02 | 0.01 |
| RS-FairFRS[20%] | Server | Learning Rate ($\alpha$) | 0.012 | 0.009 |
| | | Regularizer ($\lambda^r$) | 0.05 | 0.02 |
| | | Fairness Regularizer ($\lambda^f$) | 1.5 | 1.25 |
| | Clients | Learning Rate ($\gamma$) | 0.5 | 0.2 |
| | | Regularizer ($\lambda^r$) | 0.01 | 0.02 |
| | | Fairness Regularizer ($\eta$) | 1.75 | 1.55 |
| RS-FairFRS[100%] | Server | Learning Rate ($\alpha$) | 0.04 | 0.15 |
| | | Regularizer ($\lambda^r$) | 0.05 | 0.01 |
| | | Fairness Regularizer ($\lambda^f$) | 2.5 | 2.0 |
| | Clients | Learning Rate ($\gamma$) | 0.65 | 0.22 |
| | | Regularizer ($\lambda^r$) | 0.01 | 0.02 |
| | | Fairness Regularizer ($\eta$) | 1.75 | 1.55 |

Table 2: Hyperparameter Values

| Datasets | Attribute | Algorithms | RMSE | RMSE on Advantaged Group | RMSE on Disadvantaged Group | Demographic Bias |
|---|---|---|---|---|---|---|
| ML1M | Age | FedRec | 0.89890 ± 0.00024 | 1.00784 ± 0.00212 | 0.89476 ± 0.00026 | 0.22409 ± 0.00432 |
| | | RS-FedRec | 0.89922 ± 0.00040 | 1.00855 ± 0.00181 | 0.89506 ± 0.00042 | 0.22479 ± 0.00405 |
| | | RS-FairFRS [20%] | **0.80581 ± 0.00165** | **0.86298 ± 0.00441** | **0.80364 ± 0.00161** | **0.10244 ± 0.00660** |
| | | RS-FairFRS [100%] | 0.90466 ± 0.00080 | 1.00881 ± 0.00084 | 0.90071 ± 0.00085 | 0.21397 ± 0.00285 |
| | Gender | FedRec | 0.89890 ± 0.00024 | 0.91902 ± 0.00101 | 0.89112 ± 0.00047 | 0.05446 ± 0.00218 |
| | | RS-FedRec | 0.89922 ± 0.00040 | 0.91923 ± 0.00045 | 0.89134 ± 0.00047 | 0.05448 ± 0.00171 |
| | | RS-FairFRS [20%] | **0.80695 ± 0.00104** | **0.81883 ± 0.00404** | **0.80365 ± 0.00652** | **0.03354 ± 0.00282** |
| | | RS-FairFRS [100%] | 0.90443 ± 0.00062 | 0.92200 ± 0.00095 | 0.89750 ± 0.00079 | 0.04826 ± 0.00271 |
| ML100k | Age | FedRec | 0.88866 ± 0.00159 | 0.95684 ± 0.00537 | 0.88452 ± 0.00156 | 0.15253 ± 0.00980 |
| | | RS-FedRec | 0.88985 ± 0.00114 | 0.95893 ± 0.00475 | 0.88565 ± 0.00116 | 0.15483 ± 0.00937 |
| | | RS-FairFRS [20%] | **0.87343 ± 0.00316** | **0.90889 ± 0.00422** | **0.87127 ± 0.00321** | **0.07833 ± 0.00596** |
| | | RS-FairFRS [100%] | 0.90292 ± 0.00188 | 0.953335 ± 0.01394 | 0.90092 ± 0.00199 | 0.11760 ± 0.00325 |
| | Gender | FedRec | 0.88866± 0.00159 | 0.93465 ± 0.00214 | 0.89249 ± 0.00098 | 0.08380 ± 0.00370 |
| | | RS-FedRec | 0.88985 ± 0.00114 | 0.93512 ± 0.00147 | 0.89315 ± 0.00094 | 0.08389 ± 0.00387 |
| | | RS-FairFRS [20%users] | **0.88824 ± 0.00275** | **0.91282 ± 0.00510** | **0.87958 ± 0.00272** | **0.06503 ± 0.01114** |
| | | RS-FairFRS [100%users] | 0.90241±0.00179 | 0.93135±0.00231 | 0.89062±0.00219 | 0.079656±0.00600 |

Table 3: Training results of 4 different algorithms on real-world datasets.

| Datasets | Attribute | Algorithms | RMSE | RMSE on Advantaged Group | RMSE on Disadvantaged Group | Demographic Bias |
|---|---|---|---|---|---|---|
| ML1M | Age | FedRec | 1.28992 ± 0.0009 | 1.34240 ± 0.00050 | 1.26220± 0.00050 | 0.45860 ± 0.00021 |
| | | RS-FedRec | 1.30870 ± 0.00844 | 1.41004 ± 0.00085 | 1.28991± 0.00122 | 0.47445 ± 0.00311 |
| | | RS-FairFRS [20%] | **1.07941 ± 0.00355** | **0.90014 ± 0.00233** | **0.88700± 0.00220** | **0.20124 ± 0.00876** |
| | | RS-FairFRS [100%] | 1.50866 ± 0.00070 | 1.55281 ± 0.00081 | 1.10155 ± 0.00080 | 0.30369 ± 0.00549 |
| | Gender | FedRec | 1.27881 ± 0.00012 | 1.29940 ± 0.00120 | 1.25531± 0.000131 | 0.09971 ± 0.00880 |
| | | RS-FedRec | 1.28808 ± 0.00043 | 1.30001 ± 0.00032 | 1.26101 ± 0.00014 | 0.09818 ± 0.00160 |
| | | RS-FairFRS [20%] | **1.00121 ± 0.00812** | **0.90144 ± 0.00555** | **0.86652 ± 0.00853** | **0.06164 ± 0.00372** |
| | | RS-FairFRS [100%] | 1.44441 ± 0.00022 | 1.29909 ± 0.00099 | 1.27713 ± 0.00069 | 0.08929 ± 0.00221 |
| ML100k | Age | FedRec | 1.33451± 0.00331 | 1.45421 ± 0.00549 | 1.38331 ± 0.00106 | 0.30133 ± 0.00860 |
| | | RS-FedRec | 1.32320 ± 0.00450 | 1.42773 ± 0.00220 | 1.37877± 0.00545 | 0.30311 ± 0.00568 |
| | | RS-FairFRS [20%] | **1.12250 ± 0.00113** | **0.98182 ± 0.00661** | **0.90973 ± 0.00435** | **0.21243 ± 0.00601** |
| | | RS-FairFRS [100%] | 1.38915 ± 0.00087 | 1.43470 ± 0.00995 | 1.38001 ± 0.00221 | 0.28823 ± 0.00221 |
| | Gender | FedRec | 1.323112 ± 0.00229 | 1.37360 ± 0.00189 | 1.31004 ± 0.00022 | 0.10119 ± 0.00283 |
| | | RS-FedRec | 1.329914 ± 0.00654 | 1.36915 ± 0.00441 | 1.30899 ± 0.00107 | 0.11218 ± 0.00471 |
| | | RS-FairFRS [20%] | **1.01877 ± 0.00334** | **0.80817 ± 0.00190** | **0.77521 ± 0.00162** | **0.08225 ± 0.00188** |
| | | RS-FairFRS [100%] | 1.39592 ± 0.00227 | 1.29280 ± 0.00779 | 1.12118 ± 0.00911 | 0.90718 ± 0.00901 |

Table 4: Testing Results of 4 different algorithms on real-world datasets

amount, we can acquire much better accuracy and demographic fairness too. Further, our results emphasize an important and interesting reasoning to improved accuracy as RS-FAIRFRS[20%] not only reduces the loss on advantaged group but also on the disadvantaged group. Our model is able to achieve this due to the uniqueness of the metric $l_{ap}$ which when added to optimization function of MF, reduces loss as well as the difference between the loss on the various attributes over each stochastic gradient descent step.

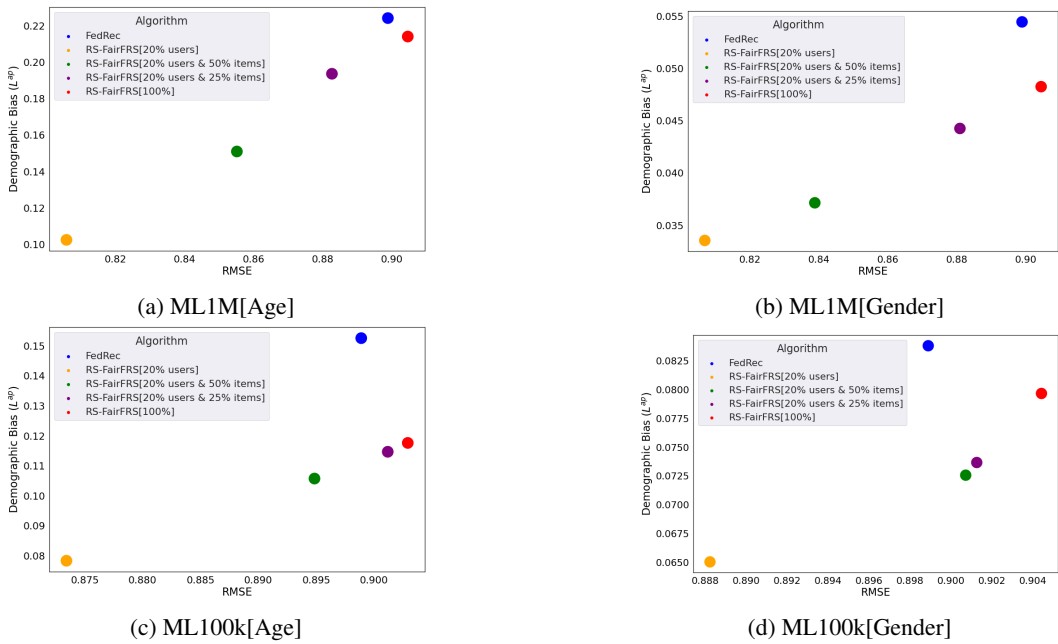

Figure 7: Fairness vs Accuracy plots

## A.6    ADDITIONAL EXPERIMENTS

This section introduces some additional findings which validate the ability of our model to improve fairness when we further reduce the amount of data on the server. Initial experiments use entire rating data from 20% users. We use the data of 20% users which are enough to represent entire local population and belong to same platform. This is a reasonable proportion to assume as atleast these many users who already use the online platform might agree to share their sensitive attributes with the server. We further toughen this assumption by considering only few of the previously rated items by these 20% users. This is done by keeping a certain proportion of items on the server. We consider 25% and 50% items of the selected 20% users. Figure 7 presents the final results obtained on these additional experiments. We plot Demographic bias vs RMSE to represent our results using scatter plot. We show the promising results of our algorithm even when dataset is further reduced on the server. The consistency of results on accuracy as well as fairness can be seen across all the datasets and sensitive attributes. Though intutive, our results suggest that having entire data of 20% users can be very conducive in obtaining a fairer as well as more accurate results. However, with only 25% of the item ratings, we are still able to obtain much reduced bias and better accuracy for all the four datasets. Moving further, if we increase data on the server by considering 50% of the item ratings, then we results which are slight better than the previous consideration of 25%. And, the best outcomes are encountered with all the item ratings.

Since RS-FAIRFRSpromotes having small amount of user's data on the server, it is evident from the results that RS-FAIRFRSwith 100% data on the server performs the worst. This happens due to the consideration of outliers which participate when all the users happen to occur on the server. Whereas, only 30% of the users from local population participate in the aggregation at server. This result emphasizes that RS-FAIRFRSneed not consider heavy fraction of users at the server. Client fraction as minimal as 20% can help in improving fairness as well as accuracy of the model.

