# OpenReview forum: "Communication Efficient Fair Federated Recommender System"
_ICLR.cc/2023/Conference — Submitted to ICLR 2023_

### Official Review · Reviewer_4W2j · 2022-10-22

**Confidence:** 3
**Correctness:** 3
**Technical Novelty And Significance:** 2
**Empirical Novelty And Significance:** 2
**Recommendation:** 5

**Clarity, Quality, Novelty And Reproducibility:**

Clarity: The writing of the draft is ok but can be further improved. For example, the terms "FairMF" and "FO-ClientBatch" suddenly appear as paragraph titles. I would recommend using one to two sentences to outline the dual update process and the terms before those paragraphs.

Quality: Good. I like the motivation of the paper as fairness is an important yet challenging topic in federated systems. The authors propose a solution with some trade-offs and the experimental results show it works.

Novelty: Low novelty. The protocol is a combination of known techniques. The proof for the sample complexity is novel but not very challenging given the strong assumptions.

Reproducibility: The code is included in the supplementary material. I did not have the chance to reproduce the results in the paper.

**Strength And Weaknesses:**

Strength:
1. The authors derive the first sample complexity for federated recommender, despite under strong assumptions.
2. The authors evaluate the proposed the method on two real-world datasets and the experimental results seem promising.

Weakness:
1. Some of the assumptions in the paper are not validated. For example, in the first paragraph of 4.1, the authors make a pretty strong assumption that the distribution of the gradients are a mixture distribution by several homogeneous distributions. I would like to see this validated on a real-world dataset.
Another assumption worth validating is ": We assume the availability of data of very few (20%) clients Dserver at the server for evaluating the fairness loss." 20% is a non-negligible percentage. Could the author give a practical scenario where such a big portion of users' data are available?
2. Algorithm 1 is too sketchy. It is understandable if the reader fully understand the context but as a pseudocode it is not self-contained.

**Summary Of The Paper:**

The paper propose a federated recommender system which guarantees fairness and communication efficiency while trying to minimize the accuracy drop.

**Summary Of The Review:**

Overall this is a borderline paper.

The quality of the paper, including the motivation, the design of the protocol, the evaluation, is okay. The writing can be further improved as I pointed out in the weakness and clarity sections.

The novelty is low. Like I mentioned in the novelty section, the proposed protocol is composed of known techniques. The math for deriving sample complexity is ok but not very challenging.

-----------------------------------------------------------------
I would like to thank the authors for their answers to my question. I decide to keep the score after reading the rebuttal.

---

> ### Author Response · Authors · 2022-11-09
> **Reply to reviewer 4W2j**
>
> 1.) **Assumption on 20% data on server:** Some past literature works like [Kanaparthy et al. (2021), Song et al. (2022)]  have considered the assumption of some data on the server to achieve fairness and personalization in federated classification tasks. Motivated by such works, we also consider the availability of a small amount of data on the server and show how it can help reduce demographic bias in federated recommendation tasks. Moreover, the small percentage of  20% user’s data on the server is very realistic as these many users can be willing to reveal their sensitive attributes in many situations, as suggested by reviewer qZGd and we have provided more details and preliminary experimental results in the rebuttal of reviewer qZGd.
>
> **Assumption about distribution of gradients as a mixture distribution by several homogeneous distribution:**  Algorithms like MF work by training in a way that they make use of latent features of the users and items making it feasible for users as well as items to be clustered into groups on the basis of item vectors. So, for recommendation datasets, assuming that some clients exhibit a rating pattern which might be followed by another client too is quite obvious. Our experiments using K-Means in the Appendix Section A.2 show that there are around 20 clusters of users. Also, in Fig 6 (a) of Appendix we illustrate that the average number of samples in such clusters generated by K-Means are almost similar in count. Based on these results, we make the assumption of users sharing similar item vectors in each cluster. Further note that *we are not clustering on the gradients of the item vectors*, but on the item vectors. The sample complexity bounds are provided on the final rating predicted by the users which uses the dot product with the user vectors (not shared) with the item vectors (similarity is assumed).
>
> 2) We thank you for the suggestions to improve our paper. We will update the main paper by adding the exact procedures of FairMF as well as ClientFilling and provide a brief introduction to these before their explanation arrives in the text.
>
> 3) **About Novelty:** We consider, for the first time, demographic bias in federated recommender systems. While demographic bias in federated classification tasks was considered in the past, we want to emphasize that federated learning in recommender systems fundamentally differs from federated learning in classification settings, as each client contains data from only one user in recommendation systems. Dual-fair update mechanism adopted by our algorithm for demographic fairness is the first of its kind. Generally,  the sensitive attributes of all the users are shared with the server to achieve fairness goals. Our technique can ensure privacy to 80% of the users (by training without using their demographics) and fair recommendations to all. Also, previous works for fair federated classification settings usually initialize fair global models and train by using fair global model updates locally. We emphasize that we design FairMF for obtaining a fair global model and then instead of using global updates of item vectors locally, we minimize the difference between the local item vectors and fair global item vectors. This is the major step which trains local clients towards fairness instead of just using fair global item vectors for local training.
>
>  In the past, many works have used the random sampling technique as a heuristic to improve communication efficiency, however, none of the existing works have provided the sample complexity bounds. Thus, the theoretical analysis for selecting an ideal fraction of clients in each communication round, and designing a dual-fair update mechanism for mitigating group bias are both novel contributions in terms of federated recommendations.

---

> ### Author Response · Authors · 2022-11-15
> **Response to reviewer 4W2j**
>
>
> We thank you for your feedback and suggestions. We have updated the main paper with suggested alterations and appendix with some additional experiments.

---

### Official Review · Reviewer_r9UQ · 2022-10-26

**Confidence:** 5
**Correctness:** 3
**Technical Novelty And Significance:** 2
**Empirical Novelty And Significance:** 2
**Recommendation:** 3

**Clarity, Quality, Novelty And Reproducibility:**

The paper is not clearly described its details. Some of key method relies on the other papers. The paper proposes source code for evaluation.


**Strength And Weaknesses:**

Strength:

1. The targeting problem is an important branch of federated learning.
2. The paper is well written.
3. The proposed method is novel.


Weaknesses:
1. The proposed method relies on two assumptions: 1) clustering clients in the recommendation system, and 2) availability of 20% data at server. These two assumptions are too strong that limits the value of the proposed method.
2. It is unclear how the clients will be clustered. Because the mentioned fairness is group-wise, the clustering result is very important to decide which client will contribute to the corresponding group.
3. In Algorithm 1, there are two subfunctions, FairMF and ClientFilling,  that need to be cleary described.
4. Some experiment results need to further analyzed. For exampmle, in Figure 3, why all curves are very unstable at the first server rounds, and the communication rounds is too few to demonstrated that the MF is converged. In Table 3, please analyse why RS-FairFRS(100%) performs worse than RS-FairFRS(20%).


**Summary Of The Paper:**

The paper proposes to improve the communication efficiency of the federated recommender system by incorporating random sampling. Specifically, the paper conducts a theoretical analysis to investigate how the reduced number of sampled clients will impact the model’s accuracy. Upon the above framework, the paper further proposes a de-biased recommender system by adding the fairness constraints in the training loss.


**Summary Of The Review:**

The paper is a combination of two mechanisms: 1) random sampling for improving efficiency, and 2) regularization term for fairness. The proposed methods relies on two impractical assumptions. Moreover, the proposed method is reuse the existing technology without limited novelty. The experimental analysis needs to be strengthened to better support its claim.

---

> ### Author Response · Authors · 2022-11-09
> **Response to reviewer r9UQ**
>
> First, we clarify clustering and data on the server assumptions. In turn, we explain the clustering assumption as only relevant to determining theoretical bounds for sampling and not to designing the dual fair update mechanism. Finally, we explain the missing details of some plots and ensure to update the same in the main paper.
>
> 1.) **Assumption Regarding Clusters:**  Our algorithm's working does not depend on the clustering assumption. Clustering is considered only for theoretical analysis of random sampling. It is used to provide the sample complexity bounds when uniform sampling is considered to reduce the communication cost. Many previous works [*Malinovsky et al. (2022), Charles et al. (2022)*] have used random sampling as a heuristic to show communication efficiency in federated learning models. However, none provides bounds on the ideal fraction of clients which should be sampled. Our main contribution lies in the fact that while bounding the sample complexity is in general hard, having a clustering assumption on the underlying data makes it possible to provide the sample complexity bounds.
>
> Algorithms like MF work by training in a way that they make use of latent features of the users and items making it feasible for users as well as items to be clustered. So, for recommendation datasets, assuming that some clients exhibit a rating pattern which might be followed by another client too is obvious. Our experiments using K-Means in the Appendix Section A.2 show that there are around 20 clusters of item vectors. Also, in Fig 6 (a) of the Appendix, we illustrate that the average number of samples in such clusters generated by K-Means is almost similar in the count. Based on these results, we make the assumption of a similar number of users in each cluster. Furthermore, this assumption is not used anywhere to prove the fairness results of our model either experimentally or theoretically. We would also like to highlight that if data is highly skewed, i.e. one cluster has really less number of users, then there is no hope of getting good bounds because random sampling might completely miss the points from this cluster and may lead to high deviation from that of the model will all the data points. Therefore, we believe this is a reasonable assumption for a starting point.
>
> **Assumption Regarding 20% Data on Server:** Some past literature works [*Kanaparthy et al. (2021), Song et al. (2022)*] have considered the assumption of some data on the server to achieve fairness and personalization in federated classification tasks. Motivated by such works, we also consider the availability of a small amount of data on the server and show how it can help reduce demographic bias in federated recommendation tasks. Moreover, the small percentage of  20% user’s data on the server is very realistic as these many users can be willing to reveal their own sensitive attributes in many situations, as suggested by reviewer qZGd and we have provided more details and preliminary experimental results in the rebuttal of reviewer qZGd.
>
> 2.) **Irrelevance of clustering assumption and fairness mitigation in RS-FairFRS:** To clearly understand the clustering assumption of our model, it is important to note that the fairness mitigation strategy used in RS-FairFRS does not rely on any clustering algorithm. We will explicitly clarify this in our main updated paper. Further, in section A.2 of the Appendix, we illustrate the results obtained by the K-Means clustering algorithm and show the existence of latent clusters among the item vectors. Private Clustering is an open challenge in federated learning, but group clustering (gender/age) is irrelevant to our assumption since clustering is not used anywhere in the Dual-Fair update technique, which mainly handles the fairness of our algorithm.
>
> 3.) We thank the reviewer for suggestions in the third and fourth comments. We have explained the working of both FairMF and FO-ClientFilling already in Section 4.2. We will update the exact procedures of both FairMF and Client Filling in our main paper. We will also update the suggested experimental analysis in the main paper and explain the same here.
> *Fluctuations in initial rounds of training:* In MF, the user, as well as item vectors, are initialized randomly. Slowly as the training proceeds towards local minima; the random vectors get trained to obtain the closest rating predictions. These predictions are the dot product of the two vectors. Thus, over some time, the curve smoothens. Since our model uses random initializations and MF at its basic level, the initial readings fluctuate.
> *Better results with lesser data on the server:* 100% data in the server does not generate fairness as good as 20% due to random sampling of clients. The sampled clients make use of the server dataset to obtain fair results. 100% of server data includes outliers, which only worsens the fairness of randomly sampled clients, causing poor fairness and accuracy.

---

> > ### Comment · Reviewer_r9UQ · 2022-12-05
> > **The assumption regarding 20% of data on the server is not reasonable.**
> >
> > Thanks for the response. I read all reviewers' comments and the authors' responses. Below is my reply to the authors' response.
> >
> > 1. The assumption regarding 20% of data on the server is not reasonable. The authors give two references that considered a similar assumption in the response. However, there is no related assumption discussion in the paper--Kanaparthy et al. (2021). Please be aware of the authority of this paper because it is not been officially published yet and it is also not produced by a world-leading research group in this domain, thus it is not a standard setting to be recognised by the community. Another mentioned paper--Song et al. (2022) can't be found in the reference list. I strongly doubt the reasonability and correctness to make such an assumption.
> > 2. Incorporating random sampling into FedRec is not a contribution. There have been some FedRec models adopting random sampling during optimization, such as FedNCF [1], FedGNN [2] and FedFast [3]. In their implementation, the sampling ratio of clients in each training round is less than 35% reported in this paper.
> > 3. There is no validation data to select the hyperparameters. Are the reported experimental results obtained with the best test results?
> > 4. Is there a risk of a data breach regarding dataset utilization? As introduced in Section 5.1 - EXPERIMENTAL SET-UP, “split each dataset randomly by keeping 20% of the data for the test set and the rest for training”. Also, there are 20% of the data reserved on the server. Then, is there any overlap between the test data and the data saved by the server?
> > 5. Why not compare the performance with baselines on the full dataset but only on the two attribute groups separately?
> > 6. The proposed model to mitigate group bias can solve limited scenarios. In both methods and experiments, the model can only deal with group bias issues with two groups, while there are more groups in practical scenarios. Can the proposed method be extended to the setting with more than two groups?
> > 7. There are also FedRec models aiming at fairness in the recommendation task, such as [4-6], and they do not make the assumption regarding data on the server. What is the difference in fairness between this paper and theirs?
> >
> >
> > [1] Perifanis, Vasileios, and Pavlos S. Efraimidis. "Federated neural collaborative filtering." Knowledge-Based Systems 242 (2022): 108441.
> > [2] Wu, Chuhan, et al. "Fedgnn: Federated graph neural network for privacy-preserving recommendation." arXiv preprint arXiv:2102.04925 (2021).
> > [3] Muhammad, Khalil, et al. "Fedfast: Going beyond average for faster training of federated recommender systems." Proceedings of the 26th ACM SIGKDD International Conference on Knowledge Discovery & Data Mining. 2020.
> > [4] Zhu, Zhitao, et al. "Cali3F: Calibrated Fast Fair Federated Recommendation System." arXiv preprint arXiv:2205.13121 (2022).
> > [5] Maeng, Kiwan, et al. "Towards Fair Federated Recommendation Learning: Characterizing the Inter-Dependence of System and Data Heterogeneity." arXiv preprint arXiv:2206.02633 (2022).
> > [6] Luo, Sichun, et al. "Towards communication efficient and fair federated personalized sequential recommendation." arXiv preprint arXiv:2208.10692 (2022).

---

> ### Author Response · Authors · 2022-11-15
> **Response to reviewer r9UQ**
>
> We are thankful to the reviewer for the valuable feedback and insightful comments. We have updated the main paper as well as appendix with suggested alterations and some additional experiments.

---

### Official Review · Reviewer_uaXN · 2022-10-27

**Confidence:** 4
**Correctness:** 4
**Technical Novelty And Significance:** 3
**Empirical Novelty And Significance:** 4
**Recommendation:** 6

**Clarity, Quality, Novelty And Reproducibility:**

Clarity: the paper is well presented.

Novelty: the proposed two techniques (client sampling and dual-fair vector update) are novel

Reproducibility: It is not difficult to reproduce the results on the two public datasets, i.e., MovieLens 100K and MovieLens 1M.


**Strength And Weaknesses:**

Strength:
1 The authors derive some complexity bound in client sampling for reducing the communication cost.
2 The authors propose a dual-fair vector update technique for improving the fairness.

Weakness:
1 The authors focus on a very specific problem, i.e. rating prediction with explicit feedback, and a specific type of method, i.e., matrix factorization.

**Summary Of The Paper:**

In this paper, the authors improve a recent federated recommendation method for rating prediction with explicit feedback, i.e., FedRec [Lin et al., 2021], by reducing the communication cost and improving the fairness.

**Summary Of The Review:**

The proposed techniques are indeed novel for the specific problem, i.e., federated recommendation with explicit feedback, and are effective as shown by the empirical studies. However, the technical contribution is not very significant since the authors only focus on a very specific problem among various more important recommendation problems, e.g., matrix factorization for recommendation with implicit feedback.

---

> ### Author Response · Authors · 2022-11-09
> **Response to reviewer uaXN**
>
> **Choice of MF-based Federated Recommendation System:** We design our algorithm based on the existing federated recommender model FedRec: Federated Recommendation With Explicit Feedback (*Lin et al. (2019)*). Though there are many other recommendation models, MF is the state-of-the-art and most frequently used model since the Netflix Prize Competition (*Gomez-Uribe & Hunz (2015)*) and has outperformed all the existing models. So, we select the model that has already proven the best. We then point out some of the drawbacks of such a model when mapped to a federated setting, such as communication cost and demographic bias and then propose a model which mitigates all such challenges or issues. Further, in our paper, we clearly cite to illustrate the usage of explicit feedback in our model.
>
> **Choice of explicit feedback:** Federated Collaborative Filtering for Privacy-Preserving Personalized Recommendation Systems (*ud din et al. (2019)*) developed the first Federated Recommendation systems that use implicit feedback. However, the authors of FedRec: Federated Recommendation With Explicit Feedback (Lin et al. (2019)) argue that there is a major privacy leakage via gradients when such a setting is used for implicit feedback. Therefore, techniques like User Averaging and Hybrid Filling were developed for better privacy preservation. Thus the usage of explicit feedback in FedRec provides enhanced privacy. Hence, we consider the Federated Recommender System with explicit feedback using MF.
> Furthermore, the usage of implicit feedback also leads to the issue of positive-unlabeled problems (Saito et al. (2019)). We believe that any recommendation model which uses factorization techniques can extend our model, and we leave this as an interesting future work. Thus, our model already uses the state-of-the-art method to achieve three significant challenges of fairness, accuracy, and communication efficiency without jeopardizing the privacy of most of the users.

---

> ### Author Response · Authors · 2022-11-15
> **Response to reviewer uaXN**
>
> We thank you for your valuable feedback and suggestions. We have updated the main paper with suggested alterations and appendix with some additional experiments.

---

### Official Review · Reviewer_qZGd · 2022-10-27

**Confidence:** 4
**Correctness:** 3
**Technical Novelty And Significance:** 2
**Empirical Novelty And Significance:** 3
**Recommendation:** 6

**Clarity, Quality, Novelty And Reproducibility:**

Clarity - Overall the paper is quite clear and easy to follow. Some interpretations were ambiguous to me, such as why their method would lead to more accurate recommendations when additional fairness constraints are imposed.
Quality - Offline experiments are conducted in a competent way.
Novelty - Techniques are modestly novel. They are not a far departure from existing techniques, but are combined in a novel and coherent way.
Reproducibility - Although the techniques are described clearly and use an open dataset, I do not believe the authors open-sourced their implementation.

**Strength And Weaknesses:**

Strengths:
1. Maintaining fairness while protecting privacy is a challenging problem, and this paper shows good progress on a real-world dataset toward achieving this aim.
2. Multi-part algorithm is intuitive and well presented.
3. Proof of affect of sampling rate on federated accuracy fills a gap in existing literature.

Weaknesses:
1. The problem of maintaining fairness and privacy is incompletely solved as their solution requires a fraction (20%) of user's to sacrifice privacy. The remaining 80% of users only enjoy partial fairness as they are only regularized by a fair model without theoretical guarantees. The authors should explain how realistically they would violate 20% of users' privacy. Perhaps this fraction is comparable to the number of users who opt out / do not opt in to privacy permissions, though this group may not be a representative sample of users overall.
2. The proof of the error bound when sampling clients is unsurprising and seems somewhat obvious. This section can perhaps be moved to an appendix. Furthermore, there is an assumption that users are uniformly distributed among clusters, but in general in fairness, groups tend to be of unequal size.
3. Evaluation is performed on two different sized MovieLens datasets, so the paper is closer to validating on just a single problem domain. The authors should consider other recommender system benchmark datasets.

**Summary Of The Paper:**

This paper presents a 3-part algorithm for an efficient and fair federated recommender system that preserves privacy. They promote privacy and efficiency by randomly sampling item gradients from only a fraction of clients. The authors present a theorem to give bounds on how the sampling rate affects accuracy. They then develop a global fair model by only using sensitive information about a client's group from a fraction of the population. The global model is then used to regularize item representations in local models to maintain fairness without exposing further sensitive information. The authors show this leads to increased model accuracy, reduced bias, and reduced communication on MovieLens datasets.

**Summary Of The Review:**

I am marginally inclined to accept this paper. The authors address an important issue with an intuitive and well-explained algorithm. They demonstrate their technique on a real-world dataset and show improvements in efficiency, fairness, and accuracy. However, their solution still manages to violate privacy for 20% of the population while providing only incomplete fairness for the remaining users. Also the novlety of their technique is somewhat modest. The authors should use the feedback from reviews to strengthen the final version of the paper.

---

> ### Author Response · Authors · 2022-11-09
> **Response to reviewer qZGd**
>
> We are thankful to you for the valuable review and suggestions.
>
> 1.) **Assumption Regarding 20% Data on Server:** Considering small dataset on the server to achieve fairness in federated classification tasks is not new [*Kanaparthy et al. (2021), Song et al. (2022)*]. Previous works, however, considered 5% of the overall data on the server. We emphasize that federated learning in recommendations fundamentally differs from classification tasks, as each client contains data from one user in *recommender systems* (RS). In contrast, each client contains data from multiple data points in classification tasks. Choosing 5% of the overall data in RSs may lead to the privacy leakage of more users. Further, this is the first work to mitigate demographic bias in FRSs, and we believe that this is a reasonable assumption to start with.
>
> In practice, servers have initial data based on some consumers' prior experiences and then try to learn a better model based on their personal experiences on the individual devices. For experiments, we chose 20% of users uniformly at random. We agree that the model may fail if the server's data does not represent the entire population. Therefore, we assume that the users who agree to keep the data on the server are uniformly distributed; hence, this proportion of users would be a good representation of the entire population.
>
> *Additional Experiment:* To contrast with the existing literature, we have performed additional experiments on ML100k. Instead of keeping all the ratings of 20% of users, we consider only 25% of the items (this corresponds to only 5% of the data on the server with fewer privacy concerns). Our results suggest that we can still reduce almost 30% of demographic bias on Age attribute (RMSE of **0.90119** and Demographic Bias of **0.11465**) and 12% on gender (RMSE of **0.90127** and Demographic Bias of **0.07366**). Vanilla Fedrec had an RMSE of 0.88866 and a Demographic bias of 0.15253 on the Age attribute and an RMSE of 0.88866 and a Demographic bias of 0.08380 on the Gender attribute. We will complete the experiments with ML1M and report all the results in the updated paper before 18 November.
>
> 2.) **Assumption Regarding Clustering:**  Many previous works [*Malinovsky et al. (2022), Charles et al. (2022)*] have used the random sampling technique as a heuristic to improve communication efficiency in federated settings without providing the sample complexity bounds on the ideal fraction of clients to be sampled. Our main contribution lies in showing that while bounding the sample complexity, in general, is a hard problem, clustering assumption on the underlying data makes it possible to provide the non-trivial bounds. Our experiments using K-Means in the Appendix A.2 show the existence of around 20 clusters on item vectors. Also, Fig 6 (a) of the Appendix illustrates that the average number of samples in such clusters generated by K-Means is almost similar in the count. Further note that clustering is done based on the similarity of item vectors instead of the demographics. Therefore, as long as we have a uniform number of users sharing similar item vectors, our proofs will go through. Without this assumption, straightforward use of Hoeffding's inequality will give trivial bounds of 100% on the sample complexity, whereas we need only 30%.
> Further, the clustering assumption is used only in the proof and is not used anywhere to prove the fairness results of our model, either experimentally or theoretically. We will explicitly clarify this in our main updated paper.
> We would also like to highlight that if data is highly skewed, i.e. one cluster has fewer users, then there is no hope of getting good bounds because random sampling might completely miss the points from this cluster and may lead to high deviation. Therefore, we believe this is a reasonable assumption for a starting point.
>
> As suggested, we will move all the proofs to the appendix, keeping only lemmas and our main theorem in the main paper.
>
> 3)**Choice of Dataset:** MovieLens is the most heavily used dataset for validating recommendation models. We used two different-sized datasets on two attributes of MovieLens to validate our model in line with various existing past works. This shows the efficacy of our approach on four datasets. First, our base model FedRec [*Lin et al. (2019)*], uses these two datasets to evaluate the accuracy of their model. Hence, for a clear and fair comparison, we also use the same datasets and show how our model is more accurate, fair and better in terms of communication efficiency. Second, most of the literature in fair recommendations use MovieLens datasets like [*Yao & Huang (2017),(Koutsopoulos & Halkid (2018))*]. While some works use synthesized datasets, we use MovieLens datasets because of the public availability and presence of two different attributes of gender and age in the same dataset to validate our model on two different sensitive attributes.

---

> > ### Author Response · Authors · 2022-11-15
> > **Response to reviewer qZGd**
> >
> > We appreciate the insightful comments and recommendations that you have provided. We have updated the paper as well as appendix with suggested alterations and some additional experiments.

---

### Decision · Program_Chairs · 2023-01-20

**Decision:**

Reject

**Justification For Why Not Higher Score:**

Please see above.

**Justification For Why Not Lower Score:**

NA

**Metareview: Summary, Strengths And Weaknesses:**

The paper studies fairness and efficiency. in the context of Federated Recommender Systems (FRS). Past literature on federated supervised learning shows that sampling improves communication efficiency without jeopardizing accuracy. The authors provide sampling complexity guarantees for this process in the context of FRS, and extend their analysis and sampling approach to address demographic bias and incorporate fairness. They show empirically their proposed method reduces communication costs while maintaining fairness and accuracy.

Reviewers found the problem interesting, but contributions to be somewhat too incremental for ICLR. Broadening the setting considered would help both w.r.t. challenge behind contributions but also the paper's scope. Presentation also needs polishing: crucial details and algorithmic implementations were unclear.